# WHAT ARE GANS USEFUL FOR?

## ABSTRACT

GANs have shown how deep neural networks can be used for generative modeling, aiming at achieving the same impact that they brought for discriminative modeling. The first results were impressive, GANs were shown to be able to generate samples in high dimensional structured spaces, like images and text, that were no copies of the training data. But generative and discriminative learning are quite different. Discriminative learning has a clear end, while generative modeling is an intermediate step to understand the data or generate hypothesis. The quality of implicit density estimation is hard to evaluate, because we cannot tell how well a data is represented by the model. How can we certainly say that a generative process is generating natural images with the same distribution as we do? In this paper, we noticed that even though GANs might not be able to generate samples from the underlying distribution (or we cannot tell at least), they are capturing some structure of the data in that high dimensional space. It is therefore needed to address how we can leverage those estimates produced by GANs in the same way we are able to use other generative modeling algorithms.

## 1 INTRODUCTION

Generative Adversarial Models (GANs) (Goodfellow et al., 2014) or, in general, implicit generative models (Mohamed & Lakshminarayanan, 2017; Kingma & Welling, 2014) are extremely appealing to natural science because they promise to be the universal simulator. If I were a genetist and had access to the genome of a few millions (billions) humans, a GAN could help me imagine any possible human. If I were a climate scientist and had access to global weather data, a GAN could envision any worldwide weather pattern. These simulators are not a goal of their own, no matter how difficult they are to build. They are tools to understand genetic disease or comprehend hurricane formation or any other particularity that we are interested in. Once, we had such a universal simulator that can generate data as nature does, we could ask any relevant question and, automatically and inexpensively have the data to validate our hypothesis.

GANs have brought deep learning to generative modeling. Deep learning has been extremely successful in discriminative modeling, because they allow learning the features together with the classifier and we do not need to rely on low-dimensional-intuitively-human-engineered features to do so. In (Vapnik, 1998) the author, when criticizing the paradigms created by parametric approach, declares that: "*To find a functional dependency from data, the statistician is able to define a set of functions, linear in their parameters, that contain a good approximation to the desired function. The number of parameters describing this set is small*". Deep Learning takes this need to define a priori this set of functions. Implicit models in discriminative learning make a lot of sense. We have a clear goal in mind, like: finding objects in an image, converting speech into text or automatic machine translation, and we have a clear metric to validate our results.

But generative modeling, unlike discriminative learning, does not have such clear goals and metrics. The generative model is always an intermediate result for solving some other problem. We want a generative model to understand the available data and be able to generate/validate hypothesis about it. Broadly, we can classify generative models in two groups: those that estimate the density of the data accurately and could potentially be used for any application; and, those that extract a representation that is actionable, even though the generative model might not fully capture the data generating process, and target a specific question about the data.

In this second group, we encounter dimensionally reduction and clustering algorithms. A representative example, among many others, could be Latent Dirichlet Allocation (LDA) (Blei et al., 2003). LDA is a generative model, that allows understanding the topics in a corpus and which topics are covered in each document, but generating a document from it, would be utterly incomprehensible. GANs clearly do not fall in this category.

GANs belong to the first category, but while non-parametric models, like histograms or kernel density estimation, suffer from slow converge to the density even in the simplest low dimensional problems (Vapnik, 1998; Lugosi & Nobel, 1996; Wasserman, 2010). GANs seem to be immune, because we have seen that with very limited data they have been able to generate compelling images (Radford et al., 2016; Salimans et al., 2016; Gulrajani et al., 2017). But, is this a proper test to assure that GANs are generating from the distribution of the data? Shouldn't we have some test in which we can say how far we are from the true underlying distribution? Shouldn't we have a specific application in mind for which GAN might be instrumental? Furthermore the visual tests that we use nowadays are clearly flawed. Because when we look from afar the 32-by-32 pixel images might look as bedrooms (Radford et al., 2016), but as soon as we magnify them and look at them one by one, we see that in most of them there is something off. For higher resolutions, the images look even more unnatural (Salimans et al., 2016). The same thing happens for text generation (Gulrajani et al., 2017).

Most of the published results for GANs have the following flavor: *GANs are unstable and we do such and such to make them easier and better to train them* (Nowozin et al., 2016; Arjovsky & Bottou, 2017; Arjovsky et al., 2017; Radford et al., 2016; Salimans et al., 2016; Gulrajani et al., 2017; Li et al., 2015; Mroueh & Sercu, 2017; Tolstikhin et al., 2017; Mescheder et al., 2017; Nagarajan & Kolter, 2017; Roth et al., 2017; Mroueh et al., 2017)[1]. We are able to generate better-looking images or even text or something else. But actually very few papers have studied if the GANs results really generate samples from the density, they just hope they do. In a way, we are still asking what can we do for GANs, instead of asking what can GANs do for us.

On one hand, our working hypothesis is that GANs will not be able to deliver accurate density estimates in high-dimensional structured spaces. On the other hand, GANs are capturing some relevant information of that data, which is providing a simulator that can still be useful even if the density estimate is wrong. There are some applications (Zhang et al., 2017; Hayes & Danezis, 2017; Hausman et al., 2017; Hitaj et al., 2017a) in which this is clearly the case. For example, GANs for password generation (Hitaj et al., 2017b) do not need accurate density estimates, because the GAN is able to capture passwords that previous methods could not. The frequencies might be off, but they are valid passwords nonetheless.

GAN-like objective functions also emerge within the context of approximate variational inference with implicit distributions (Mescheder et al., 2017; Huszár, Ferenc, 2017). Many applications leverage on this kind of highly-scalable approximate inference machinery to train interpretable latent variable models in which an accurate estimate of the underlying distribution is not mandatory, see for instance (Srivastava & Sutton, 2017; Yang et al., 2017). The unanswered relevant question is when a GAN can be safely used in an application that requires some form of density estimation. We expect that the answer to this application would be application dependent. It can take the form of a flawed density estimate or rely on some statistics of the models that match those in the real data or described a interpretable model.

In this paper, to foster the discussion about where GANs should lead us, we first examine the convergence proofs for GANs and why we believe they are incomplete or not suitable for analyzing GANs. We also revisit the papers that have look at GANs density estimate and why most of them are critical of GANs. In the second part, we show two examples that reinforce this idea that GANs density estimates are very poor and they do not generate from the same distribution the data is coming from. Our experiments concur with the papers detailed in the second section of the paper.

---

[1]The activity in this area is very large and we might have missed some relevant papers

## 2 DENSITY ESTIMATION FOR GANS

### 2.1 PROOFS OF CONVERGENCE

There are four papers in which the convergence of the GANs to the true density is analyzed. The first proof that GANs can converge to the true density was given in the GAN original paper by Goodfellow et al. (2014). Recently, a new convergence proof has been reported in (Tolstikhin et al., 2017; Liu et al., 2017). On the contrary, in (Arora et al., 2017), the authors show that there exist equilibrium points during the training of the GAN for which the generative neural net distribution would not converge to the real distribution of the data.

The analysis in the original paper (Goodfellow et al., 2014) proved that, if we knew the underlying density model, the equilibrium of the game would be a generator that would draw samples from the original distribution and a discriminator that would not be able to tell the difference. It assumed that if the discriminator has finite (VC) dimension and the number of samples tends to infinity, then the convergence will happen. But this derivation has several missing steps. First, the law of large numbers is not enough for that convergence to take place, since one will need to prove uniform convergence too (Schölkopf & Smola, 2002). Second, we need to prove that the iterative procedure, between the discriminator and generator, does not get stuck in local minima in which the generator only mimics, at best, part of the distribution. Third, we also need to analyze the convergence when the neural network grows and so does the input dimension (e.g. noise distribution). We need to sort out all of them, before we can claim that GANs converge to the distribution, because otherwise we might find that we are dropping modes or not estimating the tails of the distribution correctly. This second problem has not even been addressed yet, as we show in the experimental section. Finally, there is no bound for finite set of samples, so we will never know how far we are from achieving our goal.

In (Tolstikhin et al., 2017), the authors propose a GAN that follows a boosting procedure. New components are added to the mixture until the original distribution is recovered and they show exponential convergence to the underlying density. Our concern with the proof is its practical applicability, as it requires that, at each step, the GAN estimated density, call it $dQ$, and the true underlying density of the data, call it $dP_d$, satisfy that $\beta dQ \leq dP_d$. However, it is indeed unknown how to design a generative network that induces a density $dQ$ that would guarantee $\beta dQ \leq dP_d$ with a non-zero $\beta$ when $dP_d$ is a high-dimensional structure generative process.

In (Arora et al., 2017) the authors prove that for the standard metrics (e.g. Shannon-Jensen divergence and Wasserstein-like integral probability metrics), the discriminator might stop discriminating before the estimated distribution converges to the density of the data. They also show a weaker result, in which convergence might happen, but the estimated density might be off, if the discriminator based on a deep neural network is not large enough (sufficient VC dimension).

Finally, in (Liu et al., 2017) the authors worry about two important aspects of GAN convergence: what how good the generative distribution approximates the real distribution; and, when does this convergence takes place. For the first question the answer is the discriminator forces some kind of moment matching between the real and fake distributions. In order to get full representation of the density we will need that the discriminator grows with the data. For the second question, they show a week convergence result. This result is somewhat complementary to Arora et al. (2017), because it indicates that the discriminator complexity needs to grow indefinitely to achieve convergence. The question that remains to be answer is the rate of convergence, as the moments need to be matched for complicated distributions might require large data and complex discriminators. So in practice, we cannot tell if the generated distribution is close enough to the distribution we are interested in.

Today's GAN's results can be explain in the light of these theoretical results. First, we are clearly matching some moments that relate to visual quality of natural images with finite size deep neural networks, but they might not be deep enough to capture all relevant moments and hence they will not be able to match the distribution in these images. This explains why low resolution images, in which are visual systems interpolates what it wants to see, the results are so appealing, while in large resolution the results are far from acceptable.

## 2.2 Convergence critiques

In two recent papers (Lopez-Paz & Oquab, 2017; Sutherland et al., 2017), the authors proposed to use two-sample tests to validate GANs density estimates. Both papers reach similar conclusions, if we compare images in pixel space, none of the generated distributions by GANs pass those tests. Basically these methods notice that there are artefacts in which the two-sample tests can lock onto to distinguish between real and fake images. In the next section we replicate these experiments and find similar conclusions.

In (Sutherland et al., 2017) the authors even replicated the experiment with MNIST in Salimans et al. (2016) in which human could not distinguish between fake and real digits and showed that for the MMD test the difference was easy, because of the artifacts and because the GANs were only showing standard digits, but the GAN was not able to generate samples from the tail of the distribution. This experiment shows the risk of using human evaluation of densities estimates, as we focus more on normalcy and under-represent the unexpected. In the next section, we show another test that illustrates that GANs might be only focusing on the modes in the data and ignoring those tails, which might be critical in applications in which the tails carry the information of the extreme events that we might be interested in (e.g. think of weird genetic diseases or extreme weather patterns).

In (Lopez-Paz & Oquab, 2017), the authors also showed that if instead comparing in the pixel space, the comparison is made in some transformed space (in their case the final layer in a Resnet structure), the fake samples and the true samples were indistinguishable. This result is quite appealing, because there are some statistics in which the GANs samples are distributed like the real samples and those statistics are sufficient for our problem in hand then we might be able to rely on GANs as a simulator. The main questions is for which statistics this happens and how broad they are.

Finally, in (Arora & Zhang, 2017; Lee et al., 2017), the authors complete their theoretical paper in Arora et al. (2017) to show that GANs might be dropping modes of the distribution by adapting the birthday paradox for images. Like images comes from a continuous space they need some discretization in order to show this mode dropping effect. In this paper, we showed that this limitation can be also analyzed when the GAN is used to generate a discrete distribution. We show that the estimated probabilities are way off and we find both mode dropping and mode enhancing to happen at the same time.

## 3 Simulations

We now provide a set of experiments to support our previous discussion. We first consider two image datasets, MNIST and CIFAR-10, and later we will revisit the recently proposed GAN-approach to enhance password guessing (PassGAN) (Hitaj et al., 2017b). While in the first case we have used the GAN implementation proposed in (Salimans et al., 2016), PassGAN has been implemented using the improved Wasserstein GAN described in (Gulrajani et al., 2017).

### 3.1 Simulations using MNIST and CIFAR databases

We consider a specific implementation of the generator and discriminator networks, both trained over the MNIST (60,000 labeled images of handwritten numbers, plus 10,000 extra images in the test set) and CIFAR-10 datasets (50,000 labeled natural images, plus 10,000 extra images in the test set). Also, we train the GAN with the ones from MNIST, i.e. 1-MNIST database, which contains 6739 training images of ones and 1134 test images.

We have reproduced the experiments and network design of (Salimans et al., 2016). More specifically, the generator network is a 4 layer deep CNN, with both batch and weight normalization (Ioffe & Szegedy, 2015). The input $\mathbf{z}$ is a 100-dimensional Gaussian. The discriminator network is a 9 layer deep convolutional network with dropout, weight normalization and a Gaussian noise layer that is added to the input. The discriminator is trained using minibatch discrimination (MD), which helps to avoid collapse of the generator by looking at multiple data examples in combination. As described in (Salimans et al., 2016), within a few iterations of stochastic gradient descend (SGD), the generator network is already able to provide visually appealing images. In all cases, we have run 600 epochs, where per iteration all data mini-batches are processed with SGD once. In Figure 1,

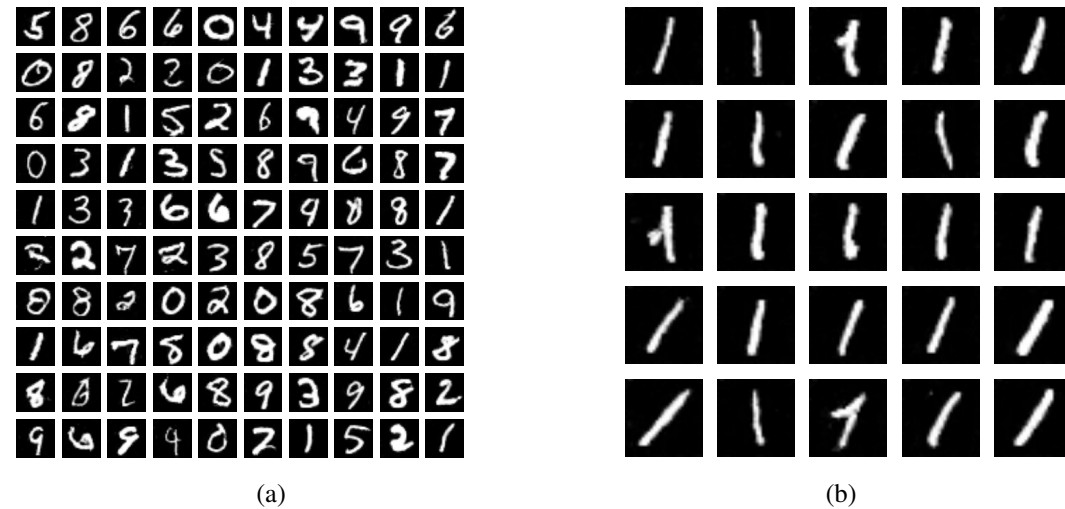

(a)              (b)

Figure 1: Samples drawn from the generator network trained over the MNIST and 1-MNIST datasets.

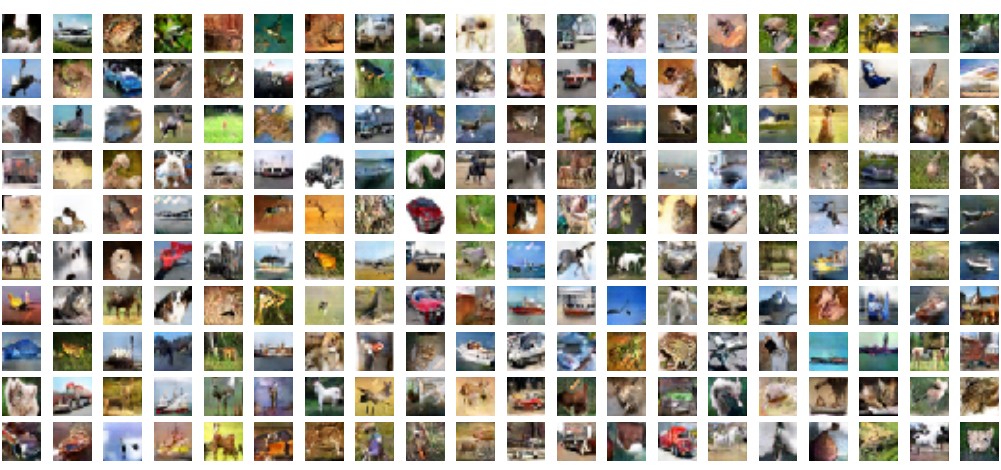

Figure 2: 200 samples drawn from the generator network trained over CIFAR-10 dataset.

we illustrate samples from the generator network trained over MNIST (a) and 1-MNIST (b), and in Figure 2 samples when it is trained over CIFAR-10. In most cases the samples are visually appealing, and it is likely that in most of them a person is not able to distinguish if they are real images or generated by our network. These results match those in Salimans et al. (2016).

In the next subsection we analyzed the Kernel Two-Sample Test results and we have left for the appendix the results in which we illustrate the mode dropping and the under sampling of the tails of the distribution.

### 3.1.1 KERNEL TWO-SAMPLE TEST

In contrast to (Salimans et al., 2016), where the inception score is introduced as a metric of the quality of the images sampled from the generator network, we propose a statistical hypothesis test to study whether the distributions $p(\mathbf{x})$ and the probability distribution of images induced by the generator network, i.e., $p_g(\mathbf{x}) = G(\mathbf{z}_i; \theta_G)$, are different. This problem is known as the Two-Sample Test (Anderson et al., 1994). We use the Kernel Two-Sample Test proposed in (Gretton et al., 2012), where the test statistic is the Maximum Mean Discrepancy (MMD), defined as the largest difference in expectations over functions in a unit ball of a characteristic reproducing kernel Hilbert space (RKHS). When MMD is large, the samples are likely from different distributions. In (Gretton et al., 2012), unbiased empirical estimators to the MMD metric are presented, and further, it is shown that the empirical MMD estimate shows a statistically significant difference between distributions. Given i.i.d. samples $X \sim p(\mathbf{x})$ and $\tilde{X} \sim p_g(\mathbf{x})$, an statistical hypothesis testing is used to distinguish between the null hypothesis $\mathcal{H}_0 : p = p_g$ and the alternative hypothesis $\mathcal{H}_a : p \neq p_g$. This is achieved by comparing the empirical MMD estimate with a particular threshold $\epsilon$: if the threshold is exceeded, then the test rejects the null hypothesis.

Given the *level* $\alpha$ of the test (an upper bound on the Type I error), we use bootstrap resampling on the aggregated data to obtain a test threshold $\epsilon$[2]. In our experiments we have used 1000 bootstrap shuffles, the radial basis function kernel with a bandwidth set as median distance between points in the aggregate sample. Also, we fix $\alpha = 0.05$. We use $N_{\text{Test}}$ samples taken at random from the MNIST and CIFAR-10 *test* sets and the same number of samples drawn from the generator network. Results are averaged over 1000 realizations. In Table 1 (a) we include the percentage of times the null hypothesis $\mathcal{H}_0 : p = p_g$ was rejected. We also include in the table the average MMD value and two times its standard deviation, as well as the estimated threshold $\epsilon$. In Table 1 (b), we randomly split the MNIST and CIFAR-10 test datasets into two disjoint sets, and we perform the same statistical test, where ideally the null hypothesis $\mathcal{H}_0$ should never be rejected. The comparison of results in Table 1 (a) and (b) leaves no doubt. As the number of data points increase, the empirical MMD gets further from the test threshold, strongly indicating a significant statistical difference between distributions. In contrast, the MMD metric computed in Table 1 (b) does not grow with $N_{\text{Test}}$ and remains very close to the threshold $\epsilon$. We can safely conclude that the GAN density model does not match the distribution of the training images. As mentioned before, the use of Two-Sample statistical hypothesis test to study the distribution induced by GAN has been also proposed in (Lopez-Paz & Oquab, 2017; Sutherland et al., 2017) with similar conclusions.

## 3.2 PASSGAN, A GENERATIVE MODEL FOR PASSWORD GUESSING

PassGAN (Hitaj et al., 2017b) is a recently-introduced technique that aims at generating passwords by extracting distribution information from password leaks. PassGAN builds upon the work of Gulrajani et al. (2017), which shows remarkably good results on GAN-based text generation.

In this section we show that, however, PassGAN experiences a severe mode-dropping effect. This is consistent with the observations presented in the previous sections of this paper. We illustrate how the distribution of PassGAN's output presents modes that do not exits in the real distribution of passwords. While this clearly suggest that PassGAN's density estimate is not correct, we show that these new modes are not completely random. Rather, it is reasonable for a user to use them as a password. This provides further evidence that correctly leveraging the generative distribution induced by the GAN beyond density estimation is still an open problem.

After training PassGAN, we used the generative network to obtain "fresh" password samples, which were then used for password guessing. Using the same network design as in Gulrajani et al. (2017), we trained a Wasserstein-GAN on a leaked portion of the RockYou dataset (RockYou, 2010) using 80% of the data for training (23,679,744 total passwords, 9,925,896 unique passwords), and the remaining 20% (5,919,936 total passwords, 3,094,199 unique passwords) as testing data. The results obtained using this process are highly competitive with state-of-the-art rule-based approaches adopted in John-the-Ripper (the Ripper, 2017) and Hashcat (HashCat, 2017) password cracking tools, as shown in (Hitaj et al., 2017b).

---

[2]See http://www.gatsby.ucl.ac.uk/~gretton/mmd/mmd.htm

| Data Set | $N_{\text{Test}}$ | % $\mathcal{H}_0$ rejected | MMD | $\epsilon$ |
|----------|-------|-------------|-----|---|
| MNIST | 100 | 84 | $1.41 \pm 0.46$ | 1.16 |
| MNIST | 1000 | 100 | $2.80 \pm 0.77$ | 1.24 |
| MNIST | 5000 | 100 | $8.96 \pm 1.45$ | 1.24 |
| CIFAR-10 | 100 | 95 | $1.76 \pm 1.12$ | 1.16 |
| CIFAR-10 | 1000 | 100 | $5.64 \pm 2.66$ | 1.13 |
| CIFAR-10 | 5000 | 100 | $45.15 \pm 7.20$ | 1.11 |

(a)

| Data Set | $N_{\text{Test}}$ | % $\mathcal{H}_0$ rejected | MMD | $\epsilon$ |
|----------|-------|-------------|-----|---|
| MNIST | 100 | 59 | $1.24 \pm 0.36$ | 1.18 |
| MNIST | 1000 | 44 | $1.25 \pm 0.41$ | 1.24 |
| MNIST | 5000 | 46 | $1.27 \pm 0.34$ | 1.24 |
| CIFAR-10 | 100 | 46 | $1.24 \pm 0.43$ | 1.18 |
| CIFAR-10 | 1000 | 58 | $1.35 \pm 0.76$ | 1.21 |
| CIFAR-10 | 5000 | 61 | $1.34 \pm 0.72$ | 1.19 |

(b)

Table 1: In (a), we use the Kernel Two-Sample Test with real images and samples drawn from the generator network. In (b), we use instead real images taken from two disjoint sets on the MNIST and CIFAR-10 test datasets.

We generated $10^{10}$ samples with PassGAN, out of which 528,834,530 (5.29%) were unique samples. These passwords matched 2,774,269 (46.9%) of the testing dataset. When matched against a leaked passwords dataset on which PassGAN was not trained (LinkedIn), the model was able to guess 4,996,980 (i.e., 11.5%) passwords out of 43,354,871 unique entries.

Table 2 shows the 30 most frequent passwords in the subset of the RockYou dataset used to train PassGAN, sorted in decreasing order based on their frequency. The second column provides the occurrence of each password in the dataset, whereas the third column provides the frequency of that password in the training data. In the fourth column, we present the frequency estimated by the GAN, which represents the number of times a certain password was generated with respect to the total number of passwords generated. The generator did not follow the underlying distribution: there are password samples that are significantly underrepresented by the GAN, and even instances of mode dropping. (See, e.g., "password", "abc123", "babygirl" and "anthony". Passwords present in the training data, but not in the GAN's output, are labeled as "not generated" in Table 2.) These results show a severe mode-dropping effect, as the 30th most common password with over 8,000 repetition in the over 23 million passwords is not generated by the GAN.

Furthermore, in Table 3, we present the 30 passwords produced by PassGAN with the highest number of occurrences. Interestingly, the model often produced passwords that have low or no representation in the training dataset, such as '123256' (appearing 6 times in the training data), 'iluv!u&' (which is not present in the training data). This shows a different effect compared to Table 2. However, PassGAN is always generating new modes that are somewhat structured, and that are not simply sequences of random letters. Passwords like "dangel", "michel", or "ilove", for example, are reasonable user-generated passwords. However, the GAN does generates them with a frequency that is too low. This shows that GANs might not be able to match all the relevant moments, even though they are still successful at capturing relevant information.

## REFERENCES

N.H. Anderson, P. Hall, and D.M. Titterington. Two-sample test statistics for measuring discrepancies between two multivariate probability density functions using kernel-based density estimates.

| Password | Occurrence in Training Data | Frequency in Training Data | GAN Estimated Frequency |
|---|---|---|---|
| 123456 | 232,844 | 0.98% | 100,971,288 (1.0%) |
| 12345 | 63,135 | 0.27% | 21,614,548 (0.22%) |
| 123456789 | 61,531 | 0.26% | 22,208,040 (0.22%) |
| password | 47,507 | 0.20% | 85,889 (8.6e-4%) |
| iloveyou | 40,037 | 0.17% | 10,056,700 (0.10%) |
| princess | 26,669 | 0.11% | 190,796 (0.0019%) |
| 1234567 | 17,399 | 0.073% | 7,545,708 (0.075%) |
| rockyou | 16,765 | 0.071% | 55,515 (5.5e-4%) |
| 12345678 | 16,536 | 0.070% | 5,070,673 (0.051%) |
| abc123 | 13,243 | 0.056% | 6,545 (6.5e-5%) |
| nicole | 12,992 | 0.055% | 206,277 (0.0021%) |
| daniel | 12,337 | 0.052% | 3,304,567 (0.033%) |
| babygirl | 12,130 | 0.051% | 13,076 (1.3e-4%) |
| monkey | 11,726 | 0.050% | 116,602 (0.0012%) |
| lovely | 11,533 | 0.049% | 1,026,362 (0.010%) |
| jessica | 11,262 | 0.048% | 220,849 (0.0022%) |
| 654321 | 11,181 | 0.047% | 19,912 (1.9e-4%) |
| michael | 11,174 | 0.047% | 517 (5.2e-6%) |
| ashley | 10,741 | 0.045% | 116,858 (0.0012%) |
| qwerty | 10,730 | 0.045% | 135,124 (0.0013%) |
| iloveu | 10,587 | 0.045% | 4,839,368 (0.048%) |
| 111111 | 10,529 | 0.044% | 101,903 (0.0010%) |
| 000000 | 10,412 | 0.044% | 108,300 (0.0011%) |
| michelle | 10,210 | 0.043% | 739,220 (0.0073%) |
| tigger | 9,381 | 0.040% | 658,360 (0.0066%) |
| sunshine | 9,252 | 0.039% | 3,628 (3.6e-5%) |
| chocolate | 9,012 | 0.038% | 12 (1.2e-7%) |
| password1 | 8,916 | 0.038% | 6,427 (6.4e-5%) |
| soccer | 8,752 | 0.037% | 25 (2.5e-7%) |
| anthony | 8,752 | 0.036% | not generated |

Table 2: Top-30 most frequent passwords present on the dataset used to train PassGAN. The first column shows passwords from the RockYou training dataset; the second column reports the number of occurrences of each password in the training dataset; the third column shows the corresponding frequency; and the forth column illustrates the frequency estimated by the GAN.

*Journal of Multivariate Analysis*, 50(1):41 – 54, 1994.

Martin Arjovsky and Leon Bottou. Towards principled methods for training generative adversarial networks. *5th International Conference on Learning Representations (ICLR)*, 2017.

Martín Arjovsky, Soumith Chintala, and Léon Bottou. Wasserstein GAN. *CoRR*, abs/1701.07875, 2017. URL http://arxiv.org/abs/1701.07875.

Sanjeev Arora and Yi Zhang. Do gans actually learn the distribution? an empirical study. *CoRR*, abs/1706.08224, 2017. URL http://arxiv.org/abs/1706.08224.

Sanjeev Arora, Rong Ge, Yingyu Liang, Tengyu Ma, and Yi Zhang. Generalization and equilibrium in generative adversarial nets (gans). In *Proceedings of the 34th International Conference on Machine Learning, ICML 2017, Sydney, NSW, Australia, 6-11 August 2017*, pp. 224–232, 2017. URL http://proceedings.mlr.press/v70/arora17a.html.

David M. Blei, Andrew Y. Ng, and Michael I. Jordan. Latent dirichlet allocation. *J. Mach. Learn. Res.*, 3:993–1022, March 2003. ISSN 1532-4435.

| GAN-generated Passwords | Occurrence in GAN | Frequency in GAN | Frequency in Training Data |
|---|---|---|---|
| 123456 | 100,971,288 | 1.01% | 232,844 (0.98%) |
| 123456789 | 22,208,040 | 0.22% | 61,531 (0.26%) |
| 12345 | 21,614,548 | 0.22% | 63,135 (0.27%) |
| iloveyou | 10,056,700 | 0.10% | 40,037 (0.17%) |
| 1234567 | 7,545,708 | 0.075% | 17,399 (0.073%) |
| angel | 6,384,511 | 0.064% | 8,425 (0.036%) |
| 12345678 | 5,070,673 | 0.051% | 16,536 (0.070%) |
| iloveu | 4,839,368 | 0.048% | 10,587 (0.045%) |
| angela | 3,377,148 | 0.034% | 4,548 (0.019%) |
| daniel | 3,304,567 | 0.033% | 12,337 (0.052%) |
| sweety | 2,560,589 | 0.026% | 5,140 (0.022%) |
| angels | 2,455,602 | 0.025% | 6,600 (0.028%) |
| maria | 1,582,718 | 0.016% | 3,178 (0.013%) |
| loveyou | 1,541,431 | 0.015% | 6,797 (0.029%) |
| andrew | 1,307,400 | 0.013% | 6,666 (0.028%) |
| 123256 | 1,294,044 | 0.013% | 6 (2.5e-5%) |
| iluv!u | 1,268,315 | 0.013% | 0 (0%) |
| dangel | 1,233,188 | 0.012% | 43 (1.8e-4%) |
| michel | 1,190,127 | 0.012% | 794 (0.0033%) |
| marie | 1,187,051 | 0.012% | 1,788 (0.0076%) |
| andres | 1,055,809 | 0.011% | 3,016 (0.013%) |
| lovely | 1,026,362 | 0.010% | 11,533 (0.049%) |
| 123458 | 989,324 | 0.010% | 181 (7.6e-4%) |
| sweet | 968,822 | 0.010% | 2,366 (0.010%) |
| prince | 920,415 | 0.0092% | 2,883 (0.012%) |
| ilove | 888,109 | 0.0089% | 555 (0.0023%) |
| hello | 861,067 | 0.0086% | 6,270 (0.026%) |
| angel1 | 840,056 | 0.0084% | 3,454 (0.015%) |
| iluveu | 826,944 | 0.0083% | 30 (1.3e-4%) |
| 723456 | 820,268 | 0.0082% | 6 (2.5e-5%) |

Table 3: Top-30 most frequent passwords produced by PassGAN after generating $10^{10}$ samples. The first column shows the passwords; the second column presents the number of occurrences of each password generated by the GAN; the third column provides the frequency of each password; and the fourth colum shows the corresponding frequency of each password in the RockYou training dataset.

Ian Goodfellow, Jean Pouget-Abadie, Mehdi Mirza, Bing Xu, David Warde-Farley, Sherjil Ozair, Aaron Courville, and Yoshua Bengio. Generative adversarial nets. In Z. Ghahramani, M. Welling, C. Cortes, N. D. Lawrence, and K. Q. Weinberger (eds.), *Advances in Neural Information Processing Systems 27*, pp. 2672–2680. Curran Associates, Inc., 2014. URL `http://papers.nips.cc/paper/5423-generative-adversarial-nets.pdf`.

A. Gretton, K. Borgwardt, M. Rasch, B. Schölkopf, and A. Smola. A kernel two-sample test. *Journal of Machine Learning Research*, 13:723–773, March 2012.

Ishaan Gulrajani, Faruk Ahmed, Martin Arjovsky, Vincent Dumoulin, and Aaron Courville. Improved training of wasserstein gans, 2017. URL `http://arxiv.org/abs/1704.00028`.

HashCat, 2017. URL `https://hashcat.net`.

Karol Hausman, Yevgen Chebotar, Stefan Schaal, Gaurav S. Sukhatme, and Joseph Lim. Multimodal imitation learning from unstructured demonstrations using generative adversarial nets. *CoRR*, abs/1705.10479, 2017. URL `http://arxiv.org/abs/1705.10479`.

Jamie Hayes and George Danezis. ste-gan-ography: Generating steganographic images via adversarial training. *CoRR*, abs/1703.00371, 2017. URL `http://arxiv.org/abs/1703.00371`.

Briland Hitaj, Giuseppe Ateniese, and Fernando Perez-Cruz. Deep models under the GAN: Information leakage from collaborative deep learning. *CCS'17*, 2017a. URL `https://arxiv.org/pdf/1702.07464.pdf`.

Briland Hitaj, Paolo Gasti, Giuseppe Ateniese, and Fernando Perez-Cruz. PassGAN: A Deep Learning Approach for Password Guessing. *CoRR*, abs/1709.00440, 2017b. URL `https://arxiv.org/pdf/1709.00440.pdf`.

Huszár, Ferenc. Variational inference using implicit distributions. *CoRR*, 2017. URL `https://arxiv.org/pdf/1702.08235.pdf`.

Sergey Ioffe and Christian Szegedy. Batch normalization: Accelerating deep network training by reducing internal covariate shift. In David Blei and Francis Bach (eds.), *Proceedings of the 32nd International Conference on Machine Learning (ICML-15)*, pp. 448–456. JMLR Workshop and Conference Proceedings, 2015. URL `http://jmlr.org/proceedings/papers/v37/ioffe15.pdf`.

Diederik P. Kingma and Max Welling. Auto-encoding variational bayes. In *International Conference on Learning Representations (ICLR)*. 2014.

Holden Lee, Rong Ge, Tengyu Ma, Andrej Risteski, and Sanjeev Arora. On the ability of neural nets to express distributions. In *Proceedings of the 30th Conference on Learning Theory, COLT 2017, Amsterdam, The Netherlands, 7-10 July 2017*, pp. 1271–1296, 2017. URL `http://proceedings.mlr.press/v65/lee17a.html`.

Yujia Li, Kevin Swersky, and Richard S. Zemel. Generative moment matching networks. In *Proceedings of the 32nd International Conference on Machine Learning, ICML 2015, Lille, France, 6-11 July 2015*, pp. 1718–1727, 2015. URL `http://jmlr.org/proceedings/papers/v37/li15.html`.

LinkedIn. Linkedin. URL `https://hashes.org/public.php`.

Shuang Liu, Olivier Bousquet, and Kamalika Chaudhuri. Approximation and convergence properties of generative adversarial learning. *CoRR*, abs/1705.08991, 2017. URL `http://arxiv.org/abs/1705.08991`.

David Lopez-Paz and Maxime Oquab. Revisiting Classifier Two-Sample Tests for GAN Evaluation and Causal Discovery. *5th International Conference on Learning Representations (ICLR)*, 2017. URL `http://arxiv.org/abs/1610.06545`.

Gabor Lugosi and Andrew Nobel. Consistency of data-driven histogram methods for density estimation and classification. *Ann. Statist.*, 24(2):687–706, 1996.

Lars Mescheder, Sebastian Nowozin, and Andreas Geiger. On distinguishability criteria for estimating generative models. *34th International Conference on Machine Learning (ICML 2017)*, 2017. URL https://arxiv.org/pdf/1701.04722.pdf.

Lars M. Mescheder, Sebastian Nowozin, and Andreas Geiger. The numerics of gans. *CoRR*, abs/1705.10461, 2017. URL http://arxiv.org/abs/1705.10461.

Shakir Mohamed and Balaji Lakshminarayanan. Learning in implicit generative models. *5th International Conference on Learning Representations (ICLR)*, 2017.

Youssef Mroueh and Tom Sercu. Fisher GAN. *CoRR*, abs/1705.09675, 2017. URL http://arxiv.org/abs/1705.09675.

Youssef Mroueh, Tom Sercu, and Vaibhava Goel. Mcgan: Mean and covariance feature matching GAN. In *Proceedings of the 34th International Conference on Machine Learning, ICML 2017, Sydney, NSW, Australia, 6-11 August 2017*, pp. 2527–2535, 2017. URL http://proceedings.mlr.press/v70/mroueh17a.html.

Vaishnavh Nagarajan and J. Zico Kolter. Gradient descent GAN optimization is locally stable. *CoRR*, abs/1706.04156, 2017. URL http://arxiv.org/abs/1706.04156.

Sebastian Nowozin, Botond Cseke, and Ryota Tomioka. f-gan: Training generative neural samplers using variational divergence minimization. pp. 271–279, 2016.

Alec Radford, Luke Metz, and Soumith Chintala. Unsupervised representation learning with deep convolutional generative adversarial networks. *4th International Conference on Learning Representations (ICLR)*, 2016.

RockYou. Rockyou, 2010. URL http://downloads.skullsecurity.org/passwords/rockyou.txt.bz2.

Kevin Roth, Aurélien Lucchi, Sebastian Nowozin, and Thomas Hofmann. Stabilizing training of generative adversarial networks through regularization. *CoRR*, abs/1705.09367, 2017. URL http://arxiv.org/abs/1705.09367.

Tim Salimans, Ian Goodfellow, Wojciech Zaremba, Vicki Cheung, Alec Radford, Xi Chen, and Xi Chen. Improved techniques for training gans. In D. D. Lee, M. Sugiyama, U. V. Luxburg, I. Guyon, and R. Garnett (eds.), *Advances in Neural Information Processing Systems 29*, pp. 2234–2242. Curran Associates, Inc., 2016. URL http://papers.nips.cc/paper/6125-improved-techniques-for-training-gans.pdf.

B. Schölkopf and A. Smola. *Learning with kernels*. M.I.T. Press, 2002.

Akash Srivastava and Charles Sutton. Autoencoding variational inference for topic models. *International Conference on Learning Representations (ICLR)*, 2017. URL https://arxiv.org/pdf/1703.01488.pdf.

Dougal J. Sutherland, Hsiao-Yu Tung, Heiko Strathmann, Soumyajit De, Aaditya Ramdas, Alex Smola, and Arthur Gretton. Generative models and model criticism via optimized maximum mean discrepancy. *International Conference on Learning Representations (ICLR)*, 2017. URL https://arxiv.org/pdf/1703.01488.pdf.

John the Ripper, 2017. URL http://www.openwall.com/john/.

Ilya O. Tolstikhin, Sylvain Gelly, Olivier Bousquet, Carl-Johann Simon-Gabriel, and Bernhard Schölkopf. Adagan: Boosting generative models. *CoRR*, abs/1701.02386, 2017. URL http://arxiv.org/abs/1701.02386.

Vladimir N. Vapnik. *Statistical Learning Theory*. Wiley, New York, 1998.

Larry Wasserman. *All of Statistics: A Concise Course in Statistical Inference*. Springer Publishing Company, Incorporated, 2010. ISBN 1441923225, 9781441923226.

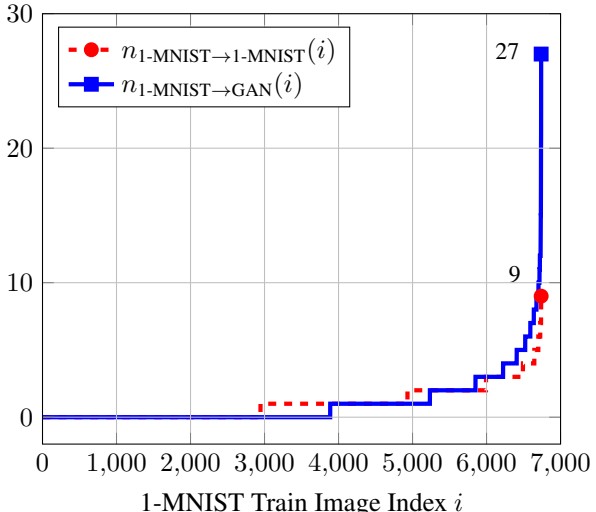

Figure 3: We represent the $n_{\text{1-MNIST}\rightarrow\text{GAN}}(i)$ and $n_{\text{1-MNIST}\rightarrow\text{1-MNIST}}(i)$ profiles computed for 6739 images sampled at random from the generator network and for the 6739 images in the 1-MNIST training set.

Zichao Yang, Zhiting Hu, Ruslan Salakhutdinov, and Taylor Berg-Kirkpatrick. Improved variational autoencoders for text modeling using dilated convolutions. *34th International Conference on Machine Learning (ICML 2017)*, 2017. URL `http://proceedings.mlr.press/v70/yang17d/yang17d.pdf`.

Yizhe Zhang, Zhe Gan, Kai Fan, Zhi Chen, Ricardo Henao, Dinghan Shen, and Lawrence Carin. Adversarial feature matching for text generation. In *Proceedings of the 34th International Conference on Machine Learning, ICML 2017, Sydney, NSW, Australia, 6-11 August 2017*, pp. 4006–4015, 2017. URL `http://proceedings.mlr.press/v70/zhang17b.html`.

## APPENDIX: EXPLAINING THE GAP: NEAREST NEIGHBOR ANALYSIS BETWEEN SAMPLE SETS

The samples generated by the GAN failed the MMD two sample test even though the generated images look as good as the original images in MNIST and CIFAR-10. We believe that the GAN fails the test because the samples are too concentrated on the modes and does not explore the full density model. This is why the samples generated for MNIST all look very good and we are almost never surprised by the weird digits that we find sometimes in the original MNIST.

We now perform a simple experiment to demonstrate the generative distribution overfit to the dominant modes of $p(\mathbf{x})$. Consider first the 1-MNIST dataset, with 6739 training images. After jointly training the discriminative and generative networks, we sample from the generator network a set of 6739 images and, for each of them, we compute the nearest neighbor (NN) in the 1-MNIST training set. We denote by $n_{\text{1-MNIST}\rightarrow\text{GAN}}(i)$ the number of times that the $i$-th image in the 1-MNIST training set is the NN of images in the fake dataset generated by the GAN. Note that large $n_{\text{1-MNIST}\rightarrow\text{GAN}}(i)$ values will correspond to images from the 1-MNIST that the GAN tends to over reproduce. The $n_{\text{1-MNIST}\rightarrow\text{GAN}}(i)$ profile is shown in Figure 3, where we have sorted the index images according to increasing values of $n_{\text{1-MNIST}\rightarrow\text{GAN}}(i)$.

We compare this result with the leave-one-out (LOO) NN profile for the 6739 images used for training, denoted by $n_{\text{1-MNIST}\rightarrow\text{1-MNIST}}(i)$. This profile is shown by a red dashed line in Figure 3. The results clearly indicate that there is a subset in the 1-MNIST training set that is over-reproduced in the generated artificial dataset, and thus, they correspond to modes in $p_g(\mathbf{x})$ that are not so-dominant in $p(\mathbf{x})$. On the other end of the spectrum the GAN represents much fewer images than the LOO NN

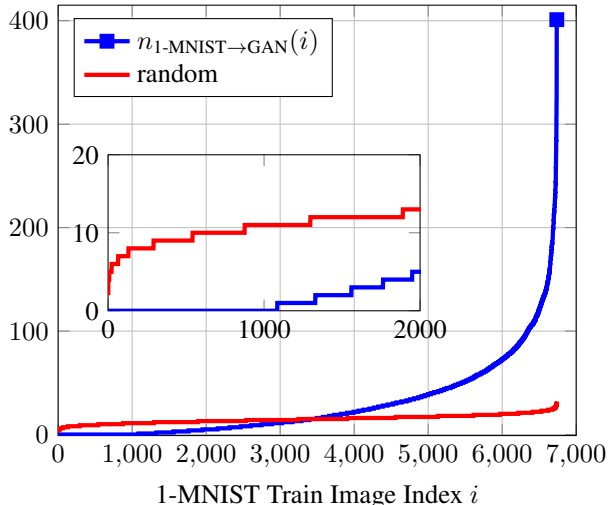

Figure 4: We represent the $n_{\text{1-MNIST}\rightarrow\text{GAN}}(i)$ profile computed for $10^5$ images sampled at random from the generator network using the two NNs of each image.

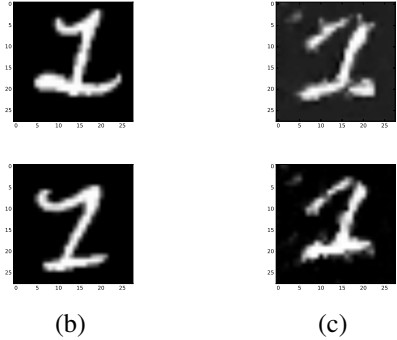

(b)                    (c)

Figure 5: In (b), we represent the two images that are furthest from their neighbors within the MNIST training set. In (b), we represent the two images that are furthest from their neighbors in artificial set of images sampled from the generative network.

profile (almost 1000 samples less). This profile leads us to believe that the GAN density estimation over represents the modes and under represents the tails of the distribution.

In Figure 4 the $n_{\text{1-MNIST}\rightarrow\text{GAN}}(i)$ profile is computed for $10^5$ generated images *and the two NNs are found for each image*. In this case, there is still 1,300 images of ones (20% of the 1-MNIST database) that do not have a first or second NNs in 100,000 GAN generated images. The image that is most popular appears as the first or second NN in 0.4% of the cases, which seems to corroborate our initial findings that the GAN over samples the modes and under sample the tails of $p(\mathbf{x})$.

An alternative illustration that reinforces the idea that the distribution $p_g(\mathbf{x})$ is indeed not representing the tails of $p(\mathbf{x})$ is provided in Figure 5, where in column (a) we represent the two ones from 1-MNIST training set furthest from their LOO NNs. In (b), we represent the two images in the fake dataset sampled from the generative network (using $10^5$ images) that are furthest from their LOO NNs. While in the ones from the real database, Figure 5(a), we observe images with marked calligraphic style (a reasonable *unlikely* one), in (b) we can appreciate images where the large distance is explained by residual noise rather than by an unusual calligraphic style. The first column of images will never be reproduced by the GAN, while the second set are noisy regular-one images.

In Figures 6 we have reproduced the same experiments for the full MNIST train set (a), where we have sample at random 60,000 images for the GAN, and the CIFAR-10 training set (b), where

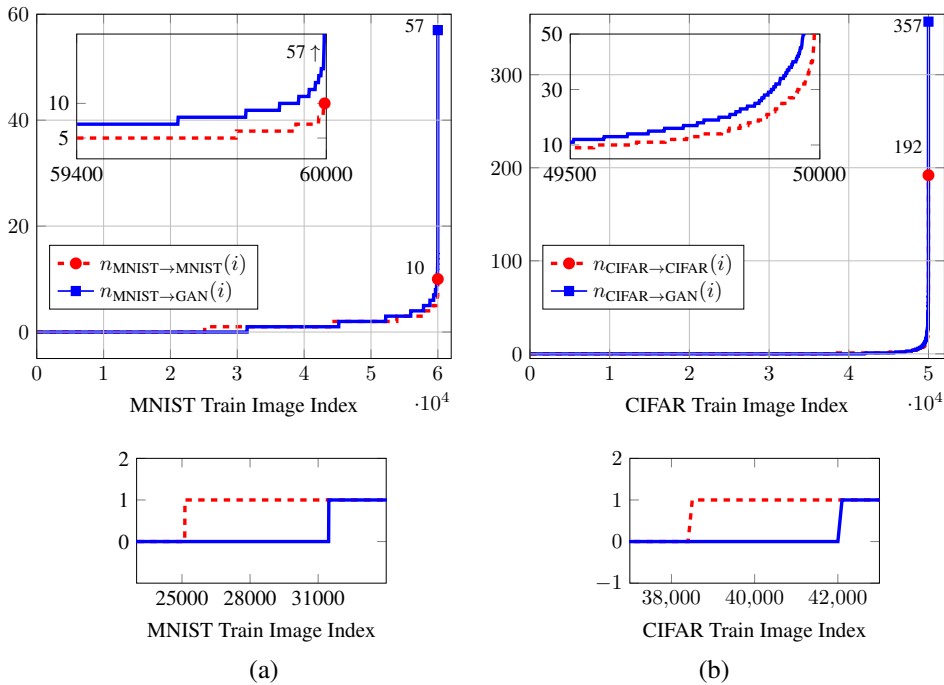

Figure 6: In (a), we represent $n_{\text{MNIST}\to\text{GAN}}(i)$ and $n_{\text{MNIST}\to\text{MNIST}}(i)$ computed for MNIST with 60,000 images. In (b), we reproduce the experiments for CIFAR with 50,000 images.

50,000 images have been sampled from the GAN. Results in both cases corroborate our conclusions, showing that the distribution $p_g(\mathbf{x})$ overrepresents certain subsets of images of the training set and under represent the tails of the distribution. For the NMIST data the 1% most popular images represent 8.32% of the images in the GAN dataset and only 5.48% in the original MNIST data, while needing 6000 less images to find the NNs for all GAN generated images. This shows that the GAN covers a significantly less fraction of the original distribution and overfits to the modes. For the CIFAR-10, the results are even more extreme.

