# OpenReview forum: "WHAT ARE GANS USEFUL FOR?"
_ICLR.cc/2018/Conference — Reject_

### Official Review · AnonReviewer2 · 2017-11-20
**Some interesting things, but not enough**

**Rating:** 3
**Confidence:** 5

**Review:**

The main take-away messages of this paper seem to be:

1. GANs don't really match the target distribution. Some previous theory supports this, and some experiments are provided here demonstrating that the failure seems to be largely in under-sampling the tails, and sometimes perhaps in introducing spurious modes.

2. Even if GANs don't exactly match the target distribution, their outputs might still be useful for some tasks.

(I wouldn't be surprised if you disagree with what the main takeaways are; I found the flow of the paper somewhat disjointed, and had something of a hard time identifying what the "point" was.)

Mode-dropping being a primary failure mode of GANs is already a fairly accepted hypothesis in the community (see, e.g. Mode Regularized GANs, Che et al ICLR 2017, among others), though some extra empirical evidence is provided here.

The second point is, in my opinion, simultaneously (i) an important point that more GAN research should take to heart, (ii) relatively obvious, and (iii) barely explored in this paper. The only example in the paper of using a GAN for something other than directly matching the target distribution is PassGAN, and even that is barely explored beyond saying that some of the spurious modes seem like reasonable-ish passwords.

Thus though this paper has some interesting aspects to it, I do not think its contributions rise to the level required for an ICLR paper.

Some more specifics:

Section 2.1 discusses four previous theoretical results about the convergence of GANs to the true density. This overview is mostly reasonable, and the discussion of Arora et al. (2017) and Liu et al. (2017) do at least vaguely support the conclusion in the last section of this paragraph. But this section is glaringly missing an important paper in this area: Arjovsky and Bottou (2017), cited here only in passing in the introduction, who proved that typical GAN architectures *cannot* exactly match the data distribution. Thus the question of metrics for convergence is of central importance, which it seems should be important to the topic of the present paper. (Figure 3 of Danihelka et al. https://arxiv.org/abs/1705.05263 gives a particularly vivid example of how optimizing different metrics can lead to very different results.) Presumably different metrics lead to models that are useful for different final tasks.

Also, although they do not quite fit into the framing of this section, Nowozin et al.'s local convergence proof and especially the convergence to a Nash equilibrium argument of Heusel et al. (NIPS 2017, https://arxiv.org/abs/1706.08500) should probably be mentioned here.

The two sample testing section of this paper, discussed in Section 2.2 and then implemented in Section 3.1.1, seems to be essentially a special case of what was previously done by Sutherland et al. (2017), except that it was run on CIFAR-10 as well. However, the bottom half of Table 1 demonstrates that something is seriously wrong with the implementation of your tests: using 1000 bootstrap samples, you should reject H_0 at approximately the nominal rate of 5%, not about 50%! To double-check, I ran a median-heuristic RBF kernel MMD myself on the MNIST test set with N_test = 100, repeating 1000 times, and rejected the null 4.8% of the time. My code is available at https://gist.github.com/anonymous/2993a16fbc28a424a0e79b1c8ff31d24 if you want to use it to help find the difference from what you did. Although Table 1 does indicate that the GAN distribution is more different from the test set than the test set is from itself, the apparent serious flaw in your procedure makes those results questionable. (Also, it seems that your entry labeled "MMD" in the table is probably n * MMD_b^2, which is what is computed by the code linked to in footnote 2.)

The appendix gives a further study of what went wrong with the MNIST GAN model, arguing based on nearest-neighbors that the GAN model is over-representing modes and under-representing the tails. This is fairly interesting; certainly more interesting than the rehash of running MMD tests on GAN outputs, in my opinion.

Minor:

In 3.1.1, you say "ideally the null hypothesis H0 should never be rejected" – it should be rejected at most an alpha portion of the time.

In the description of section 3.2, you should clarify whether the train-test split was done such that unique passwords were assigned to a single fold or not: did 123456 appear in both folds? (It is not entirely clear whether it should or not; both schemes have possible advantages for evaluation.)

---

### Official Review · AnonReviewer3 · 2017-11-26
**limited contributions**

**Rating:** 3
**Confidence:** 4

**Review:**

This paper considers the question of how well GANs capture the true data distribution. The train GAN models on MNIST, CIFAR and a pass word dataset and then use two-kernel ample tests to assess how well the models have modeled the data distribution. They find that in most cases GANs don't match the true distribution.

It is unclear to me what the contribution of this paper is. The authors appear to simple perform experiments done elsewhere in different papers. I have not learned anything new by reading this work.  Neither the method nor the results are novel contributions to the study of GANs.

The paper is also written in a very informal manner with several typos throughout. I would recommend the authors try to rewrite the work as perhaps more of a literature review + throughout experimentations of GAN evaluation techniques. In its current form I don't think it should be accepted.

Additional comments:
- The authors claim GANs are able to perform well even when data is limited. Could the authors provide some examples to back up this claim. As far as I understand GANs require lots of data to properly train.
- on page 3 the authors claim that using human assessments of GAN generated images is bad because humans have a hard time performing the density estimation (they might ignore tails of the distribution for example) .. I think this is missing up a bunch of different ideas.. First, a key questions is *what do we want our GANs for?* Density estimation is only one of those answers. If the goal is density estimation then of course human evaluation is an inappropriate measure of performance. But if the goal is realistic synthesis of thats then human perceptual measures are more appropriate. Using humans can be ban in other ways of course since they would have a hard time assessing generalizability (i.e. you could just sample training images and humans would think the samples looked great!).

---

### Official Review · AnonReviewer1 · 2017-11-27
**Offical review for "WHAT ARE GANS USEFUL FOR?"**

**Rating:** 3
**Confidence:** 5

**Review:**

This paper tried to tell us something else about GANs except for their implicit generation power. The conclusion is GANs can capture some structure of the data in high dimensional space.

To me, the paper seems a survey paper instead of a research one.  The introduction part described the involving of generative models and some related work about GANs. However, the author did not claim what the main contributions are. Even in Section 2, I can see nothing new but all the others' work. The experimental section included some simulation results, which are weird for me since they are not quite related to previous content. Moreover, the 3.1.1 "KERNEL TWO-SAMPLE TEST" is something which has been done in other paper [Li et al., 2017, Guo et al., 2017].

It is suggested that the author should delete some of the parts describing their work and make clear claims about the main contributions of the paper. Meanwhile, the experimental results should support the claims.

---

### Decision · Program_Chairs · 2018-01-29
**ICLR 2018 Conference Acceptance Decision**

**Decision:**

Reject

**Comment:**

As the reviewers said, it is unclear what the main contribution of the paper is.